# In the absence of extensive initial training, cleaner wrasse *Labroides dimidiatus* fail a transitive inference task

**Leonore Bonin**◯*, **Redouan Bshary**

Behavioural Ecology Laboratory, Biology Institute, Faculty of Science, University of Neuchâtel, Neuchâtel, Switzerland

* leonore.bonin@unine.ch

**Data Availability Statement:** All the data and R codes are publicly accessible on figshare (https://figshare.com/projects/No_evidence_for_transitive_Inference_in_cleaner_wrasse_data/158132).

## Abstract

Transitive inference (TI) is a reasoning capacity that allows individuals to deduce unknown pair relationships from previous knowledge of other pair relationships. Its occurrence in a wide range of animals, including insects, has been linked to their ecological needs. Thus, TI should be absent in species that do not rely on such inferences in their natural lives. We hypothesized that the latter applies to the cleaner wrasse *Labroides dimidiatus* and tested this with 19 individuals using a five-term series (A > B > C > D > E) experiment. Cleaners first learned to prefer a food-rewarding plate (+) over a non-rewarding plate (-) in four plate pairs that imply a hierarchy from plate A to plate E (A+B-, B+C-, C+D-, D+E-), with the learning order counterbalanced between subjects. We then tested for spontaneous preferences in the unknown pairs BD (transitive inference task) and AE (as a control for anchors), interspersed between trials involving a mix of all known adjacent pairs. The cleaners systematically preferred A over E and showed good performance for A+B- and D+E- trials. Conversely, cleaners did not prefer B over D. These results were unaffected by the reinforcement history, but the order of learning of the different pairs of plates had a main impact on the remembrance of the initial training pairs. Overall, cleaners performed randomly in B+C- and C+D- trials. Thus, a memory constraint may have prevented subjects from applying TI. Indeed, a parallel study on cleaner wrasse provided positive evidence for TI but was achieved following extensive training on the non-adjacent pairs which may have over-ridden the ecological relevance of the task.

## Introduction

Animal cognition involves a suite of mechanisms through which an individual acquires, stores, and uses information from the environment [1]. Many learned abilities among animals may be explained with Pavlovian or operant conditioning, where individuals either learn to associate a primarily involuntary action with an external stimulus [2] or associate a voluntary action with a consequence [3]. However, it remains a major challenge to identify more complex cognitive processes, such as transitive inference (TI), in non-human animals. TI is defined as the ability to deduce the relative value of two objects that were never observed together—and thus

**Funding:** Founder: The Swiss National Science Foundation The recipient: Professor Redouan Bshary. Grant numbers: 310030B_173334 and 310030_192673 website: https://snf.ch/en/FKhU9kAtfXx7w9AI/page/home The funders had no role in study design, data collection and analysis, decision to publish, or preparation of the manuscript.

**Competing interests:** The authors have declared that no competing interests exist.

not subject to potential reinforcement learning—from other previously learned pair relationships. For instance, in a list of three objects: A, B, and C; if A is greater than B, and B is greater than C, one can deduce that A is greater than C. Despite its supposed cognitive complexity, evidence of TI has been suggested in a wide variety of non-human vertebrates, including birds (e.g., jays [4, 5]; crows [6, 7]; pigeons [8, 9]; geese [10]), mammals (e.g., chimpanzees [11]; macaques [12]; lemurs [13]; mice [14], and fish (e.g., brook trout [15]; cichlids [16]; cleaner wrasse [17]), as well as some invertebrates (e.g., paper wasps [18]). The occurrence of this capacity has been largely linked to specific aspects of social structure [4, 5, 19]. For example, in large and complex social groups, TI may be used to minimize competitive interactions and thus the cost of fighting and injuries [16, 20]. This is achieved because TI allows an individual to deduce dominance relationships from observations and avoid fighting stronger individuals.

The ecological approach to cognition [1, 21, 22] predicts a tight link between ecological needs and performance in cognitive tasks. The positive results from studies involving social dominances in fishes [15, 16] and paper wasps [18] support the ecological approach to cognition by showing a link between the ecological need and the presence of TI in species that lack highly derived brain structures like a mammalian neocortex or an avian neostriatum. More recently, intraspecific differences observed in TI capacity that are directly linked to sex and rank further demonstrate the important modulation of TI by social need [23]. The ecological approach to cognition also predicts that in the absence of a need to represent hierarchical orders in the brain, TI should be absent, again irrespective of brain anatomy. This is exemplified by a study yielding negative results for honeybees [24] compared to positive results in paper wasps [18]. However, few negative results are ever reported, and hence more studies on TI are needed on species where the ecological approach predicts failure.

Here, we tested the presence or absence of TI in the bluestreak cleaner wrasse *Labroides dimidiatus*. This protogynous species has a size-based social hierarchy [25], where the largest individual in a group is a male that has a harem of smaller females. Males have a higher reproductive output than females, and sex change is socially suppressed via aggression [25] from the male to the targeted female. While this has yet to be studied in female–female cleaner wrasse interactions, size-based hierarchies are apparently stabilized by individuals suppressing the growth of subordinate individuals through the threat of aggression [26, 27]. Cleaner wrasse harems are of small size (typically 3–6 adult females in [25] (mean 3.5 females per harem [28]). Thus, cleaner wrasse moving between harems can use absolute information (size) to assess the relative strength of a few conspecifics without the need for TI. Similarly, interactions with clients allow direct assessment of client size, mucus quality, and parasite load, so that TI is not needed to establish a hierarchy of client quality as a food patch. Therefore, if the presence of TI is driven by ecological needs [1, 21, 22], we hypothesize that we should not find it in cleaners. Note that our interpretation of the cleaner wrasse social system differs from that of colleagues who independently and almost in parallel to us, also studied TI in cleaner wrasse [17]. While we do not see an ecological need for TI in cleaner wrasse, they have a complex interspecific social life, with about 2000 interactions per day with 'client' fishes [29]. Clients visit to have ectoparasites removed, but cleaners prefer to eat client mucus [30]. The resulting conflict of interest has apparently led to the evolution of high strategic sophistication in cleaners [30, 31] allowing them to match or even outperform mammals in various tasks [31–33]. Also, cleaners have an impressive cognitive tool kit for an ectotherm vertebrate that includes generalized rule learning [34], long-term memory retention of single events [35], and mirror self-recognition [36, 37]. Thus, we decided to test a "smart" species that we predict, based on our understanding of its ecological needs, should not possess TI mechanism. This will help elucidate whether TI could be an emergent property of a complex social life (see [38] for the similarities between

intra- and interspecific social interactions) that involves cooperation, defection, and competition and thus warrants social competence [39].

To test for the presence of TI in cleaner wrasse, we used a standard experimental setup that is based on a five-term series test where the individual must learn and understand the transitive link between five objects (e.g., A > B > C > D > E). Previous studies investigating TI varied the number of objects from three to seven [18, 39, 40]. Three objects (A > B > C, testing AC pair [40]) are not sufficient for assessing TI presence/absence with this paradigm, since success in the AC test can be explained as a simple associative learning component (i.e., A was always rewarded and C never, so I choose A). Even with five objects and the crucial test B vs D, researchers have reported various potential alternative explanations for transitive-like responses [41, 42]. These explanations are linked to different types of learning, such as reinforcement history, value transfer mechanisms, and absolute knowledge. For example, TI experiments may yield a systematic difference in reinforcement history between BC and DE combinations. This is because B is losing against the most dominant object (A) and its value needs to be re-established, while E is never rewarded and hence learning D > E is relatively straightforward and achievable in fewer trials than B > C. These differences between pairs may cause individuals to favor B over D simply because B has been rewarded more often until the learning criterion had been reached, without any implication of transitive inference. To control for these alternative explanations/confounding variables, we explicitly tested for effects of the order of pair presentations and the number of trials for each object pair on individual performance in the experimental trials. Note that at the time of data collection, we were unaware that the parallel study obtained positive evidence for TI, using the same general paradigm (see below) but different training methods [17]. In light of diverging results, we dedicate an important part of the discussion to compare the methods.

## Material and methods

The study was conducted in Australia, at the Lizard Island Research Station, from February to March 2020. Due to the loss of one individual during the training phase, a total of 19 fish were tested. All individuals were released at the site of capture after the experiment. The manipulations were approved by Animal Ethics Committee (AEC), permit number CA 2019/11/1336.

### Training

The training phase aimed to make fish understand the relative value of each plate compared to the others. To do this, 4 pairs of plates (hereafter referred to as the training pairs) were presented: A+B-, B+C-, C+D, D+E-, where plus and minus indicate the reinforced and the non-reinforced choice, respectively. On the reinforced plate, a small food item of mashed prawn was put on the back of the plate. The non-reinforced plate did not contain any food. Cleaners had to choose between the two simultaneously presented plates, which were separated by an opaque barrier (Fig 1). A choice was scored once the cleaner went behind one of the plates. The cleaners could hence always inspect the back of the chosen plate, which showed the same color pattern as the front to allow a post-choice feedback. Thus, they could eat the food item if the choice had been correct, and afterward still access the non-rewarding plate to see that there was no food. In contrast, the correct plate was removed and hence inaccessible if the choice had been wrong. In order to prevent any systematic preferences in color patterns and/or biases due to the sequence of presentation of pairs of plates, we counterbalanced these variables across subjects (Fig 2).

To facilitate learning, subjects could inspect both plates during the first five trials for each pair of plates, irrespective of their initial choice. With this addition, we ensured early exposure

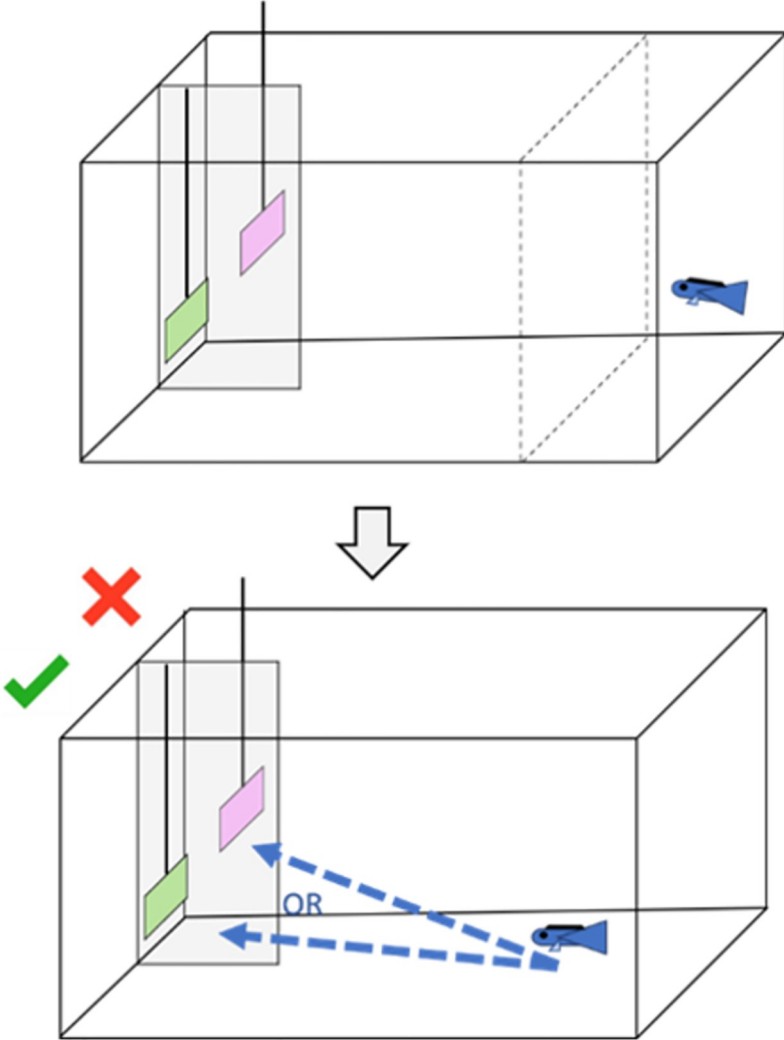

**Fig 1. Experimental setup.** The fish is put behind a see-through barrier (dotted lines) while the plates are inserted and separated by an opaque barrier (full line and grey rectangle). If the fish goes to the right one, he gets the reward and can inspect the other plate (green "V") whereas both were removed directly if he chose the wrong one (red cross). After the training, we performed the transitive inference task to test whether fish can infer the value of plates from new combinations (i.e. test combinations): B+D- and A+E-. During the task, the training combinations were randomly exposed to the fish and the test combinations were also randomly presented once in a while.

to a rewarding plate that had just before been the non-rewarding plate in the previous combination. Subjects were considered to have learned the correct choice if they reached one of the following criteria: 3 times at least 7 correct choices out of 10 trials, 2 times at least 8 correct choices out of 10 trials, or 10 correct out of 10 trials. Once subjects had subsequently reached learning criteria for all four combinations, we ran a single set of four trials in which each pair was presented once. We conducted 10 trials per fish per day.

## Transitive inference task

Prior to the task, four reminding trials were performed in order to expose individuals to each training pair again right before the task. For the task, we tested the subjects' spontaneous preferences when confronted with two novel combinations: plate A versus plate E, and plate B

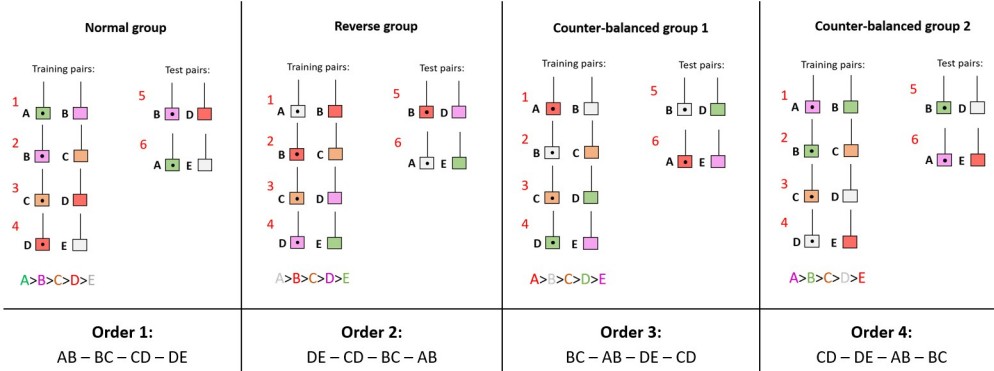

**Fig 2. Representation of the four experimental colors and order of learning groups.** The 20 initial fish were equally divided between four experimental groups that differed with respect to the color rank of the plates: Normal, reverse, counter-balanced 1 and 2. Within each group, training pairs are learned in 4 different orders. Each color group included the 4 order of learning groups. After one loss, the counter-balanced group 2 had only 4 fish and the order 1 was not represented anymore.

versus plate D. As A had previously only been used as a rewarding stimulus, and E only as a non-rewarding stimulus, we expected a clear preference for plate A when presented with E. In contrast, both plate B and plate D had been rewarding in one plate combination, and non-rewarding in the second plate combination. Thus, only if subjects had formed a hierarchical representation of the five plates, the use of transitive inference would make them spontaneously prefer plate B over plate D. Otherwise, plate choice was expected to be random. Subjects were tested with both AE and BD combinations three times each. To avoid habituation to such conditions, the six test trials were interspersed in between a sequence of learned plate pairs (AB, BC, CD, DE) where the correct choice yielded the standard reward of a prawn item. More precisely, a single test trial was included in a sequence of 4 training pairs trials. Thus, a total of 30 trials were conducted to obtain our six experimental trials. Moreover, the plate pair presented in the trial before a test was varied as much as possible within individuals and counterbalanced across individuals.

## Statistical analyses

All the analyses were carried out on R v.4.0.2.

To check whether the fish performed above chance for the entire set of training pairs during the task, we first calculated the proportion of successful trials for each individual. The % values were then used for a generalized linear mixed effect model that compared observed values to the expected binomial distribution (50% success) based on the null hypothesis that no learning had taken place. We included the fish ID as random variable to avoid pseudo-replication. The significance of the intercept was the indicator of the overall learning of the training pairs or not.

While every fish passed the learning criterion for every training pair and was hence ready for testing, we also analyzed whether all the training pairs were equally remembered during the actual TI task. We performed a Friedman rank sum test on the count of success per training pair, considering the fish identity. We then conducted pairwise comparisons between all training pairs using the Wilcoxon rank sum test with continuity correction. We built different binomial generalized linear mixed-effects models to test the different possible explanatory variables. The choice at each trial was the response variable, and the training pair was always included. Because the amount of data was not sufficient to include both the color groups and

the order of learning groups plus all the interaction terms, we had to perform separate models. After model selection using variances analyses, and because of the experimental design, the order of learning groups as well as the interaction term with the training pair were kept in the model while the color group was not.

Finally, to check which training pair was remembered above chance, we performed individual Wilcoxon rank sum tests with continuity corrections for each training pair after graphically checking that the data were homogeneously distributed around the median. The same analyses were performed on each training pair for each order of learning and repeated separately with the fish having an overall memory above 65% on the training pairs.

To compare the test pairs AE and BD, we started by investigating the data of the first trial of the task only, using a two-tailed binomial test for each pairing. When considering the three trials of the task, we applied the same method as we used on the training pairs to see whether the fish succeeded above chance and to investigate the potential roles of the different variables. In this case, the final model included only the test pair as an explanatory variable, since the amount of data did not allow to run the model including the variables considered in our other analyses (i.e., color and order of learning).

To evaluate the extent to which reinforcement history during the initial training phase may have affected choices in the BD test pair, we calculated the ratio between the number of times that B or D plate was rewarded versus non-rewarded: for B the number of succeeded BC trials over failed AB trials, for D the number of succeeded DE trials over failed CD trials. The difference between these "Rb" and "Rd" was calculated and analyzed in a non-parametric Kruskal-Wallis test (because of the small sample size) to see whether there was a systematic difference in reinforcement history between B and D, or a difference between each group of learning order. Furthermore, to test whether variation in the ratio difference could be correlated to a difference in individual success in the BD task, we used a Spearman correlation test on the ratio difference and the counts of success for BD.

All the data and R codes are publicly accessible on figshare (https://figshare.com/projects/No_evidence_for_transitive_Inference_in_cleaner_wrasse_data/158132).

## Results

### Training pairs

Overall, the fish remembered the training pairs above chance, succeeding on average in 61.4% of trials (null binomial glmer, intercept: p < 0.0001). However, the training pairs were not all equally remembered (Friedman chi-squared = 10.34, df = 3, p = 0.02, Fig 3A). AB (median = 5 with se = 0.359) was remembered significantly better than BC (median = 2 with se = 0.413, pairwise Wilcoxon rank sum test with Bonferroni correction, p = 0.008), and DE (median = 4 with se = 0.350) tended to be remembered better than BC (p = 0.07). The anchoring pairs (AB and DE) were remembered above chance levels (Wilcoxon signed rank test with continuity correction, respectively V = 143 with p = 0.001 and V = 117 with p = 0.01, Fig 3A), while the performances in trials involving the two middle pairs (BC and CD) did not differ significantly from chance (respectively V = 50.5 with p = 0.37 and V = 72 with p = 0.51). A closer inspection of the most successful fish (overall performance above 65%, 6 individuals) showed similar tendencies, with the middle pairs (BC and CD) not equally remembered (Friedman rank sum test, chi-squared = 11.509 of 3 df, p = 0.009). Due to the limited sample size of this subset of fish, the assumption for the Wilcoxon rank sum tests was not met (homogeneity around the median) and no further analyses were conducted.

(a)

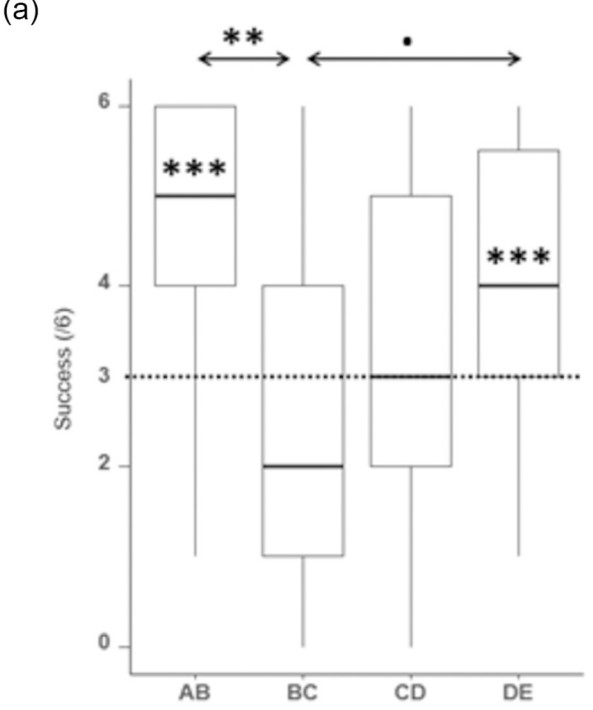

(b)

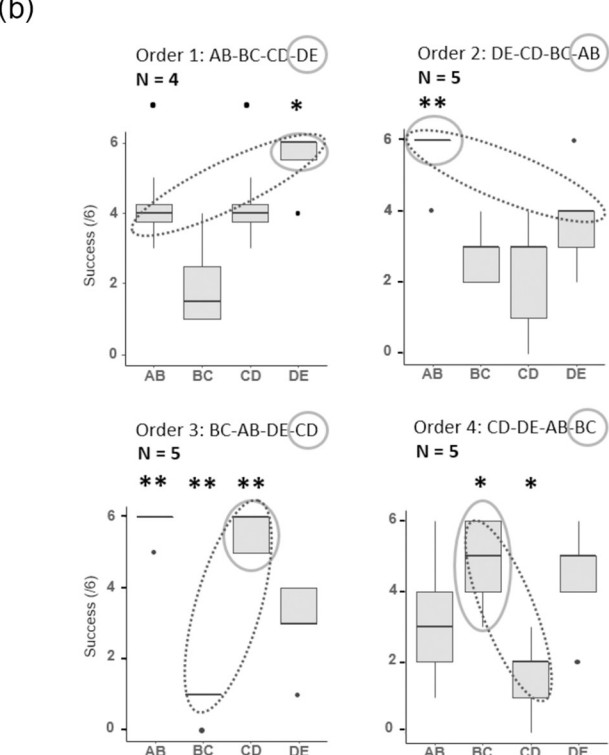

**Fig 3. Graphical representations of the count of choices for each training pair globally (A) and for each order of learning separately (B) during the task.** The A boxplot shows the number of correct choices fish made during the task. Each test pair was presented 6 times. The large horizontal lines indicate the medians, boxes are limited by the upper and the lower quartiles, and the vertical lines indicate the deciles. On the A graph, the large black dotted line indicates the random percentage of success of 50%. The stars inside the boxes show the significance of the difference

from the random threshold. Significance code: 0 '***' 0.001 '**' 0.01 '*' 0.05 '.' 0.1. The B boxplots contain the same information for each order of learning separately. In addition, the clear grey full-lined circles point the pair lastly learned and the darker dotted circles show the group of the pairs learned in the first and the last position.

We also found that there was a significant effect of learning order on remembrance of the training pairs during the task (anova on binomial glmer, Chisq = 103.8 of 9 df, p < 0.005, Fig 3B). All groups were able to reach the chance level and above for AB and DE (mostly at chance for DE), while middle pairs (BC and CD) were only remembered above chance when learned in the last position. For example, BC was the last pair to be learned for group 4 and they remembered it in 80% of trials (Wilcoxon rank sum test, W = 22.5, p = 0.006, Fig 3B), whereas the three other groups show performances at or lower than chance level.

## Test pairs

For the AE pair on the first trial of the task only, 19 of 19 subjects chose the A plate (two-sided binomial test: probability of success = 1 and p < 0.0001, Fig 4A). In contrast, only 8 of 19 cleaners chose B over D (two-sided binomial test, probability of success = 0.42 and p = 0.65, Fig 4A).

To compare our results to the study of Hotta et al. 2020, we checked whether the fish who learned the training pair in the rank-descending order (from highest to lowest) as their subjects had done (our order 1: AB, BC, CD, DE) would show similar performances. Among these 4 individuals, two chose B during the first trial, compared to 4 of 4 in their study (Note that Hotta et al. erroneously reported 3 of 4 correct first choices; the archived data show 4 of 4).

Considering all 3 trials per task, the results remain unchanged. For the AE pair, fish significantly preferred plate A over plate E (mean of success = 2.6, Wilcoxon signed rank test with continuity correction, V = 186.5, p < 0.001), while they did not solve the BD pair above chance (mean of success = 1.47, V = 92, p = 0.92). The four fish trained in a similar order to those from Hotta et al. (2020) made five correct choices out of twelve trials. This strongly contrasts with the findings of Hotta et al. (2020), which reported a strong preference for D (our B) in the BD pair in all subjects, with a minimum of 10 of 12 correct choices per fish.

Because of the difference among groups of learning order for the training pairs, even if the learning order appeared non-significant overall in the analyses of the test pairs, we checked for the group that remembered the training pair BC the most (73.3% of all trials were correct during the task). It is also the group that learned BC last. In this group of fish (N = 5, learning order 4), 3 of 5 fish chose B over D in the 3 trials. The 2 last fish succeeded only in the last trial and one of these successfully remembered the training pairs 42% of the time (highlighting a more general weakness in the remembrance of the pairs). Note also that DE was equally remembered in this group (around 75%, does not appear significantly different from the random threshold, probably due to the small sample size). Due to the small sample size (N = 5), no subsequent analyses were carried out for these groups.

In our final analysis, we evaluated the reinforcement history of plates B and D. The differences between Rb and Rd ratios were not similar among the four groups of learning order (Kruskal-Wallis, Chisq = 9.95, p = 0.02, Fig 5). The first group significantly differed from group 2 and 4 (pairwise comparisons using Fisher's least significant difference, for both: difference = 10.5, p = 0.01, Fig 5) and tended to differ from group 3 in the same way (Fisher's least significant difference, difference = 7.5, p = 0.06, Fig 5).

Finally, the individual differences in the reinforcement history of B and D did not correlate with the success in the BD task (Spearman's correlation test, rho = -0.05, p = 0.8).

(a)

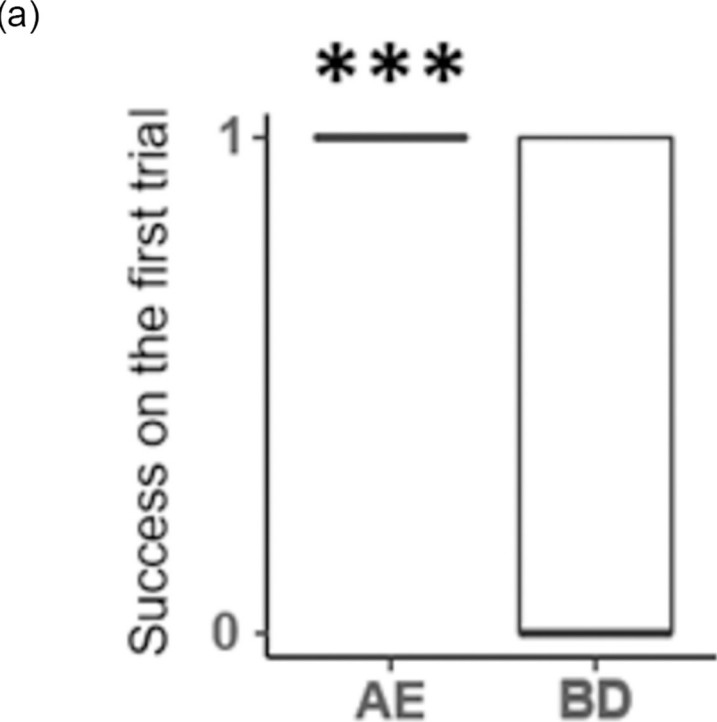

(b)

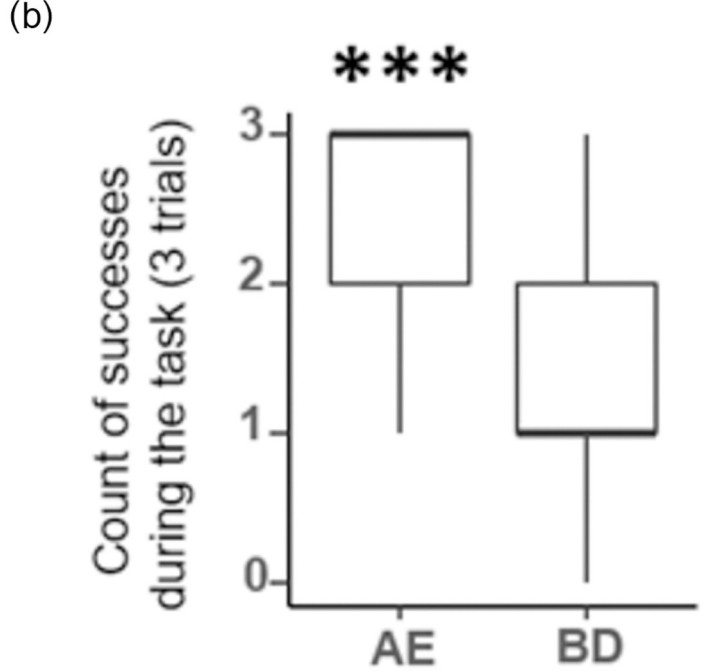

**Fig 4. Graphical representations of the success of all fish for each test pair separately including on the first trial of the task (A) and including the 3 trials (B).** The stars above the boxes show the significance of the difference from it. The large horizontal lines indicate the medians, boxes are limited by the upper and the lower quartiles and the vertical lines indicate the deciles. Significance code: 0 '***' 0.001 '**' 0.01 '*' 0.05 '.' 0.1 ' ' 1.

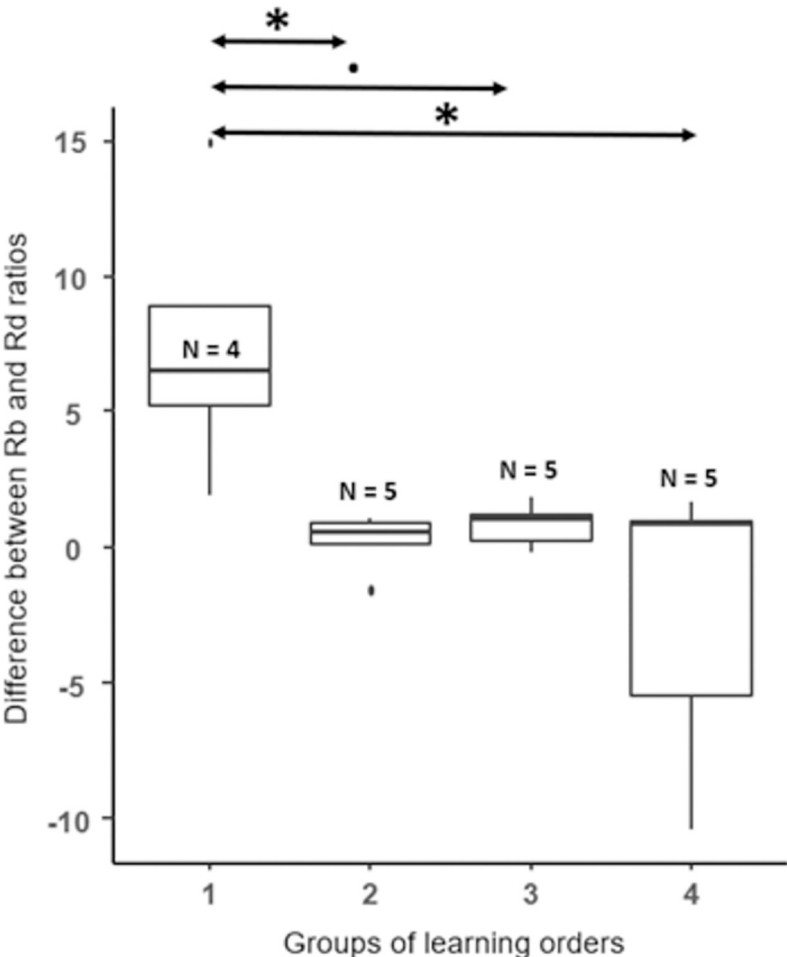

**Fig 5. Graphical representations of the difference between Rb and Rd ratios in each of the four groups of learning orders.** Rb and Rd are the ratio between the number of times that B or D were rewarded and non-rewarded during the training. The Y axis is the calculated difference between those 2 ratios. The large horizontal lines indicate the medians, boxes are limited by the upper and the lower quartiles and the vertical lines indicate the deciles. N indicates the number of fish for each group. The stars above the boxes show the significance of the comparison between the boxes indicated by the arrows. Significance code: 0 '***' 0.001 '**' 0.01 '*' 0.05 '.' 0.1.

## Discussion

An increasing number of studies have found evidence for TI among birds (e.g., jays [4, 5]; crows [6, 7]; pigeons [8, 9]; geese [10]), mammals (e.g., chimpanzees [11]; macaques [12]; lemurs [13]; mice [14]), and fishes (e.g., brook trout [15]; cichlids [16]; cleaner wrasses [17]) that is predicted to derive from the ecological needs of the species of interest. However, potential alternative explanations associated with positive findings, as well as a paucity of published negative results (but see [24]), make it important to also test species where the ecological approach to cognition predicts that they should lack TI. In this study, we tested whether cleaner fish use TI to organize information in their brain. We hypothesized that cleaners are lacking ecological conditions where TI would be a useful cognitive tool and thus should lack this cognitive ability according to the ecological approach to cognition. Our view contrasts with that of Hotta et al. [17], who proposed that the ecology of cleaner wrasse predicts the presence of TI. Readers need to decide whether they would expect TI to be present in a

protogynous species living in harems that very rarely contain more than six females, with a size-based hierarchy. While they found positive evidence for TI in cleaner fish in their almost simultaneously conducted study [17], we found no evidence for it in our study. A large part of our discussion therefore focuses on potential explanations for the diverging results. As we see it, differences in training provide the most likely explanation.

Importantly, failure to show TI in an experiment could be due to a number of factors, which may or may not be mutually exclusive. One potential explanation would be that the species tested lacks the cognitive process. Alternatively, an inability to memorize the relationship in every training pair may prevent the subject from exhibiting their capacity for TI. The latter is evident in our results, as cleaners lacked the remembrance of the middle pairs BC and CD that were presented between the actual TI trials. This was not expected, since all fish successfully reached the learning criterion during initial training for each pair. Thus, cleaners could not properly remember all four pairs. Instead, our results highlight two well-known effects in TI studies that contribute to this middle pairs outcome: (1) the recency effect (i.e., an item is favored if it was seen just before) and (2) the anchoring effect (i.e., the first and the last pair of the list are better remembered than the middle pairs). The memory performances on our training pairs were highly impacted by the recency effect, as the better-remembered pair was the one that was learned the last. Moreover, the observed U-shaped pattern of the remembrance of each training pair, in addition to the failure to remember the middle pairs above chance, are indicative of the anchoring effect. This U-shaped pattern has been previously reported in both unsuccessful [24] and successful [43–46] TI experiments. In previous successful studies [43–46], the middle pairs were still remembered above chance, unlike in the present study. The U-shaped pattern provides a potential explanation for lower preferences in the BD pair compared to the AE pair [47] and could be explained by the serial position effect (SPE), which predicts that performances on further objects in a series are higher compared to the performances on neighboring objects [41, 43, 44, 48–50]. In our case, the U-shaped pattern could not be found when the results from the 4 groups of learning order were looked at independently. Only the combination of the results from all of these groups showed the U-shaped pattern. When we looked at each order of learning independently, we could see again the anchoring effect and clearly observe the recency effect. Indeed, by comparing orders of learning 1 and 2 to orders 3 and 4, we saw that the difference between the first and last learned pairs was reduced when those were the anchoring pairs AB and DE. We observed that the better-remembered training pair for each group of learning order was the pair that was learned the last, which is a clear sign of the recency effect. These effects had the same importance for the 6 fish with an overall memory equal to or above 65%, which indicates that the failure in BC and CD was not influenced by the overall success in the memorization of the training pairs. These memory differences between the anchoring and the middle pairs can explain both the precision with which subjects preferred to approach A over E and the lack of preference in BD trials.

Several other potentially confounding variables reported in previous studies were absent in our study, and our sample size allowed us to control for different factors that could have biased the results. Our color groups avoided a particular value attribution to a specific plate (as in [16] with artificial dominance). Also, using 4 groups of counterbalanced learning order made it unnecessary to have a "bias reversal procedure" that consists of over-reinforcing D over B. Then, based on reinforcement history, the individual would significantly choose D in the BD task [6, 7]. Hotta et al. (2020) [17] used the same sequence of training for their four subjects. Our group of four subjects that experienced the same order of learning as the fish in the study by Hotta et al. [17] still showed considerably different results. Moreover, in our study, the pair of plates presented in the trial before a test was varied as much as possible within individuals and counterbalanced across individuals to avoid a systematic bias that could result from a

recency effect [5], which could, for example, influence an individual to choose B over D if exposed to B+C- in the previous trial. Furthermore, it appears that the success/failure of our fish was not affected by differences in the reinforcement history of the various plate combinations, which have been suggested as means to yield transitive-like responses in TI tasks [8, 51]. In contrast, reinforcement history could explain the success of three out of four fish used by Hotta et al. [17]. In summary, it appears that no systematic bias could have produced positive results with our study design; poor memorization of the middle pairs (BC and CD) emerges as the main reason for the cleaners' poor performance. The reason for this failure is still interesting as it sticks to ecological relevancy, so we discuss it later.

## Comparison to Hotta et al. 2020

When comparing our study to the work of Hotta et al. [17], there are several methodological differences that may have contributed to our divergent findings (presented in Table 1). One potential explanation, which we consider unlikely, is that the positive results reported by Hotta et al. [17] were due to a combination of chance and small sample size. The authors only tested four cleaners compared to our 19, but all these four cleaners consistently preferred the B plate over the D plate from trial 1 onwards over a series of 12 trials (note that the authors erroneously report only 3 of 4 correct first choices; their published data show 4 of 4 correct choices). Thus, while their sample size was much smaller than ours, we consider it parsimonious to conclude that the differences in performance are due to differences in methods.

A major difference between our two studies involves the number of training pairs. Hotta et. al [17], as well as other studies [10, 12], added non-adjacent pairs to their training (i.e., AC and

**Table 1. Main differences in methods and results between Hotta et al. 2020 and the present study.**

| Difference | | Hotta et al. 2020 | This study |
|---|---|---|---|
| Methods | Plates | Different colors | Colors and pattern differences |
| | Reward | Smeared prawn on the front, get one if succeeded or none if failed | One prawn item hidden in the back, get it if succeeded or none if failed |
| | Time between trials | 1 or 2min between the trials, 2 or 3h between the sessions | 20 to 50min between trials |
| | Sample size | N = 4 | N = 19 |
| | Training | • Phase 1:<br>A-B+, B-C+, A-C+, C-D+, D-E+, C-E+<br>Criteria: 2 x 5 or 6/6<br>• Phase 2:<br>A-B+, B-C+, C-D+, D-E+<br>Criteria: 1 x 5 or 6/6<br>• Phase 3:<br>Sessions of 8 trials, each pair presented twice. Criteria: at least 1/2 for each pair per session and over 6/8 in two consecutive sessions | • 5 trials to present each new pair.<br>• Learning:<br>A+B-, B+C-, C+D-, D+E-<br>Criteria: 3 x 7/10; 2 x 8,9 or 10/10, 9 or 10/10 + 7/10<br>• Reminding trials:<br>4 trials, each pair presented once before testing |
| | Test | 3 x 4 trials for BD | 3 x BD and 3 x AE, randomly presented within training pairs trials [10 trials per day, 1 test pair presented per 5 trials] |
| | Counterbalancing | BD colors | Colors, order of learning, which training plate was presented prior to a test pair |
| Results | Training pairs | All remembered equally | Not remembered equally |
| | Test | B-D+ succeeded | B+D- failed. |
| | | | A+E- succeeded |
| | Reinforcement history | Could explain the transitive response for 3 out of 4 fish | No global difference for B and D but variations depending on the group of learning order. Not correlated with BD success |

Note: the list is not exhaustive, only the bigger ones were listed. Readers are advised to refer to Hotta et al. 2020 to get more details on terminology or else.

CE). These combinations may have helped to consolidate the subjects' memory of the BC and CD combinations. One obvious prerequisite for the ability to demonstrate TI is that subjects memorize the ranks within all pairs so that they can deduce the ranking within novel pairs. In our series of 30 experimental trials, this prerequisite was fulfilled only for the anchoring pairs AB and DE. By adding those non-adjacent pairs, subjects obtained complete information about the rank position of plate C; it is non-rewarding when paired with A or B, and it is rewarding when paired with D and E. This extended knowledge could be necessary for the fish to establish the link between the different plates and be able to infer the value of B over D, potentially by helping to the establishment of a hierarchical list of the plates. An individual then has absolute knowledge about the plates located in both parts of the final list (i.e., A, B, C and C, D, E) that they can use to infer the new BD relationship. Without this supplementary knowledge, it is possible that the list could not be established, so the different pairs would remain as unlinked objects, preventing TI to be applied. The fact that some studies have yielded positive results using a similar experimental design to that used here—without the non-adjacent pairs training—could be a sign that the establishment of the list is already a cognitively demanding step that necessitates extended training for fish compared to some other species.

An interesting middle ground between these two cleaner studies could be to present only the four adjacent pairs but with additional rounds of BC and CD trials until all correct choices are properly memorized. On the other hand, we think that it is an important result of our study that cleaners find it challenging to memorize the middle pairs. This result raises an interesting question—what ecological conditions would yield enough training for memorization of frequently occurring pairs before TI can be used to spontaneously resolve a unique new combination? The advantage of TI is that it is a capacity that allows one to get maximum information with minimum learning. Our negative results suggest that cleaners cannot apply TI without extensive prior training in the four combinations, which negates its major advantage. While the positive results by Hotta et al. [17] seem to suggest that cleaners can apply TI provided that they receive extensive training, we reiterate that we consider their additional training on the AC and CE combinations as a potentially confounding factor. In any case, we consider the fact that our cleaners failed to remember all combinations after they all succeeded the initial learning also points at the ecological irrelevancy of the TI task. Nevertheless, we also acknowledge that replicating the social paradigm used in other species of fishes [15, 16] could help to clarify whether cleaners could succeed more easily (i.e., without an additional non-adjacent pair training) in this type of TI task compared to that utilized in a foraging context. The cognitive process of TI might still be present in cleaners, even if our study did not find evidence for it in this context.

## Reinforcement history

It has been noted several times in the literature that a difference in reinforcement history could lead to a transitive-like response in TI tasks [8, 51]. For example, fish subjected to five-item tasks in our study might on average need more reinforcement trials to learn BC than DE because B only loses against a plate with a pure positive association (plate A), while D only wins against a plate with a pure negative association (plate E). More reinforcement trials on B could thus potentially lead to a preference for B over D. That said, we did not find such inequality in the number of reinforcement trials (except for one group of learning order) in our study. Moreover, individual differences in reinforcement history ratios did not correlate with a preference for B or D.

## Transitive inference tests

The types of evidence needed to adequately test for the presence of TI continue to be debated [42, 52, 53], but the generally accepted paradigm typically involves information on social dominance—where few observations of pairs fighting may reveal information of relative strength without subjects receiving any rewards [15, 16]. In contrast, our paradigm can be criticized on the basis that a plate's 'state' is 1 (food) or 0 (no food) depending on the plate it is paired with. Such a presentation does not match on some continuous scale. Unfortunately, experiments involving reinforcement learning cannot use a continuous scale (like fighting ability) as subjects would obtain absolute information on food reward (e.g., A offers more food than B, which offers more food than C, etc.) so that TI is not needed for subjects to prefer B over D. Nevertheless, we note that the food paradigm has been used on various species ([9, 11, 24, 54]; mix with social hierarchy: [13]), often yielding positive results. These differences are interesting and interpretable. The comparison between paper wasps and bees [18, 24], as well as within-species variations in chicks [23], fits the ecological approach to cognition, and we argue that our negative results do so as well. Cleaner fish obtain absolute information about client value as a food source in nature, and their social hierarchies in each harem are based on relative body size (i.e., also based on absolute information). Thus, we would argue that cleaners should not need to employ transitive inference under natural conditions.

## Conclusion and outlook

The ecological approach to cognition has helped to identify various high performances in species that are phylogenetically distant from humans, as demonstrated in specific tasks that correspond to relevant ecological pressures. A classic example involves the selective spatial memory abilities of Clarke's nutcrackers compared to closely related species that rely less on seed caching [55]. The ecological approach has also helped to identify suitable experimental paradigms to test for advanced cognitive processes, for example by using food caching to test for episodic-like memory in scrub-jays [56]. The cleaner wrasse *L. dimidiatus* provides another suitable species for the ecological approach as their foraging conflict with clients has apparently selected for the ability to adjust to context-specific situations—i.e., whether the interaction is observed [57], whether they clean alone or with a partner [58], their own physiological state [59], client species features [60] or fish densities [33, 35]. This fine-tuned strategic sophistication raises questions regarding the underlying cognitive processes that can explain these behaviors. As it stands, cleaners perform well in tasks testing for two executive functions, namely self-control [32] and flexibility [61]. While these functions are clearly useful for their interactions with clients, transitive inference is not needed as far as we can judge. Given this, we would argue that designing studies where failure is predicted is an important complement to test the ecological approach to cognition. This is because the ecological approach implicitly assumes that brains function in rather modular ways. Thus, positive results where the ecological approach predicts failure would identify aspects of brain functioning that are more general-purpose tools. Our study suggests that cleaner fish would not use the general-purpose tool of transitive inference in their daily life, though our data do not allow us to distinguish whether this is due to memory constraints or to the absence of the TI process. We also propose that our data can be reconciled with the results of Hotta et al. [17] in that only additional training and performance control before the crucial test may yield positive results, albeit with the caveat that associative learning cannot be excluded as a simpler mechanism. It would be interesting to test larger-brained endotherm vertebrate species that also do not need TI in their daily lives to see whether this cognitive tool might eventually emerge as a by-product of larger computing power.

## Acknowledgments

The staff from the Lizard Island Research Station. N. Truskanov and Z. Yanai for helping with the setup and manipulations, the statistician R. Slobodeanu for his advice and answers, and C. Clements for his review of the manuscript.

## Author Contributions

**Conceptualization:** Leonore Bonin, Redouan Bshary.

**Data curation:** Leonore Bonin.

**Formal analysis:** Leonore Bonin.

**Funding acquisition:** Leonore Bonin, Redouan Bshary.

**Investigation:** Leonore Bonin.

**Methodology:** Leonore Bonin, Redouan Bshary.

**Project administration:** Leonore Bonin.

**Resources:** Redouan Bshary.

**Supervision:** Redouan Bshary.

**Validation:** Leonore Bonin, Redouan Bshary.

**Visualization:** Leonore Bonin.

**Writing – original draft:** Leonore Bonin.

**Writing – review & editing:** Leonore Bonin, Redouan Bshary.

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
