## [Decision Letter · Decision Letter 0]

21 Feb 2023

PONE-D-23-02374No evidence for transitive inference in cleaner wrasse *Labroides dimidiatus**PLOS ONE*

*Dear Dr. *Bonin,

*Thank you for submitting your manuscript to PLOS ONE. After careful consideration, we feel that it has merit but does not fully meet PLOS ONE’s publication criteria as it currently stands. Therefore, we invite you to submit a revised version of the manuscript that addresses the points raised during the review process.*

*In particular, please carefully address the issues raised by Reviewer 2, including the overall framing and motivation of the manuscript, and Reviewer 1's point regarding the need for increased emphasis on the possibility that the fish failure reflects that of training, rather than inference.*

*Please submit your revised manuscript by *Apr 07 2023 11:59PM*. If you will need more time than this to complete your revisions, please reply to this message or contact the journal office at plosone@plos.org. *

*Please include the following items when submitting your revised manuscript:*

*A rebuttal letter that responds to each point raised by the academic editor and reviewer(s). You should upload this letter as a separate file labeled 'Response to Reviewers'.*

*A marked-up copy of your manuscript that highlights changes made to the original version. You should upload this as a separate file labeled 'Revised Manuscript with Track Changes'.*

*An unmarked version of your revised paper without tracked changes. You should upload this as a separate file labeled 'Manuscript'.*

**

*We look forward to receiving your revised manuscript.*

*Kind regards,*

*Rachael Miller (Harrison)*

Academic Editor

*PLOS ONE*

**

Reviewers' comments:

*Reviewer's Responses to Questions*

*

**Comments to the Author**
*

1. Is the manuscript technically sound, and do the data support the conclusions?

*The manuscript must describe a technically sound piece of scientific research with data that supports the conclusions. Experiments must have been conducted rigorously, with appropriate controls, replication, and sample sizes. The conclusions must be drawn appropriately based on the data presented. *

*Reviewer #1: Yes*

*Reviewer #2: No*

*2. Has the statistical analysis been performed appropriately and rigorously? *

*Reviewer #1: Yes*

*Reviewer #2: Yes*

*3. Have the authors made all data underlying the findings in their manuscript fully available?*

*The PLOS Data policy requires authors to make all data underlying the findings described in their manuscript fully available without restriction, with rare exception (please refer to the Data Availability Statement in the manuscript PDF file). The data should be provided as part of the manuscript or its supporting information, or deposited to a public repository. For example, in addition to summary statistics, the data points behind means, medians and variance measures should be available. If there are restrictions on publicly sharing data—e.g. participant privacy or use of data from a third party—those must be specified.*

*Reviewer #1: Yes*

*Reviewer #2: Yes*

*4. Is the manuscript presented in an intelligible fashion and written in standard English?*

*PLOS ONE does not copyedit accepted manuscripts, so the language in submitted articles must be clear, correct, and unambiguous. Any typographical or grammatical errors should be corrected at revision, so please note any specific errors here.*

*Reviewer #1: Yes*

*Reviewer #2: Yes*

*5. Review Comments to the Author*

*Please use the space provided to explain your answers to the questions above. You may also include additional comments for the author, including concerns about dual publication, research ethics, or publication ethics. (Please upload your review as an attachment if it exceeds 20,000 characters)*

*Reviewer #1: I believe this is a very good paper, well-written, clear, with properly analyzed results. It seems to me that the authors should stress more perhaps that fish showed a failure of training more than a failure of transfer (inference), i.e. that the latter is mostly the outcome of the former. At any event, I believe this paper can be published as it stands.*

*Reviewer #2: The authors set out to test transitive inference (TI) in cleaner wrasse which are known for their social cognitive complexity. They reason the task is not ecologically relevant, so even though the fish are smart in other contexts, this particular task is irrelevant to them. The reasoning is sound with the exception of a previous paper on the same species and methodology which not only found evidence of TI but also justified why they expect this species to be capable of doing it on ecological grounds. So which expectation is true?? Moreover, TI is typically examined in a social context (agonistic interactions) but here the methodology is a foraging task. It is not immediately apparent if it is appropriate to expect that TI is transferable across contexts. At least some of the introduction needs to address “the cleaner fish in the closet”. Lastly, the test seems to be set up in a discrete way, whereas my intuition says that TI would best work with continuous variables.*

The authors seem to be claiming that they found no evidence for TI, but this is not really a true picture of the results. In fact, the fish basically failed the pre-training prerequisites and should never have been tested in the TI test at all. In other words, they were destined to fail. This is particularly problematic because the middle training pairs which are essential to the ultimate TI test were simply not learned above random. The authors mention this in the discussion, but ultimately it’s a fatal problem.

Having said that, I commend the authors for so thoroughly investigating their results, because it clearly highlights potential issues with how these tests are conducted. For me this is a major strength of the paper (lessons on what not to do). Both anchoring and recency effects were present in the training data sets.

I did not find the motivation for this study to be terribly convincing. It is just not clean. One paper says TI is to be expected given the fish’s ecology. Here the authors say its not. Would it not be simpler to just revisit Hotta and say you’ve greatly increased the sample size? That sounds like a better justification to me.

So overall my feeling is the motivation for the study needs to change, as does the take-home message: They failed because we failed… lessons learned.

Specific comments:

Abstract: It is a bit odd that the opening statement suggest that you do no expect to find that cleaner wrasse are capable of AI given their ecology but your last sentence says there is already evidence of TI in this species. This is a little confusing.

L60: evidence of TI has been

L67: I guess the question here is, does it make sense to test TI outside of social hierarchies? The other cleaner fish paper did as I recall but the other fish ones used agonistic interactions.

L70: This is fine, but 2 of the 3 studies done on fish support the ecological explanation. The other doesn’t, although the introduction of Hotta et al 2020 claims their task is ecologically relevant. Here you claim it isn’t. So who to believe??

L86: growth through the threat of aggression

L87: So here you are suggesting they don’t need TI because size is a true predictor of hierarchy position. Do you know if the hierarchy in your model is truly linear? How do you then explain the previous result which used pretty much the same approach? Moreover, im not sure you can transpose TI from a social context to a feeding context. This is an interesting question in its own right.

L135: I have niggling doubts about this task. In my mind TI would be particularly useful for determining where things lie on a sliding scale, A is bigger than B which is bigger than C. But here the procedure is a little bit abstract. In some cases B has a reward (B+C1), in others it doesn’t (A+B-). It’s quite complicated.

L137: in previous iterations of this test the “non preferred” plate would have flake food. Why no reward as apposed to a non-preferred award?

L204: between the number of times that

L228L sooo the fish failed to learn the middle associations? Is there any point in testing for TI given this result? Especially since your test involved contrasts with D

L235: Again this is a problem since the outcome of the TI will be dependent on what came before it.

L262: earned BC last

L281: Yeah im not sure you can sell this idea too hard given Hora et al’s results. What you could say is you have repeated that test with greater sample size. I think this would be a more convincing motivation for the study.

L282: organize information in their

L287: My feeling is that your training regime failed, so the test for TI was not valid. You must seriously consider this as the main reason for your negative finding.

L296: (45) and unsuccessful

L356 & L381: see my earlier comments about this being a discreet test as opposed to a continuous test.

L389: except they don’t because your training failed. This is likely a false negative.

*L409: I also think expectations of failure are valuable in the literature, but beware false negatives.*

*6. PLOS authors have the option to publish the peer review history of their article (what does this mean?). If published, this will include your full peer review and any attached files.*

**

*Reviewer #1: No*

*Reviewer #2: **Yes: **Culum Brown*

**

*While revising your submission, please upload your figure files to the Preflight Analysis and Conversion Engine (PACE) digital diagnostic tool, https://pacev2.apexcovantage.com/. PACE helps ensure that figures meet PLOS requirements. To use PACE, you must first register as a user. Registration is free. Then, login and navigate to the UPLOAD tab, where you will find detailed instructions on how to use the tool. If you encounter any issues or have any questions when using PACE, please email PLOS at figures@plos.org. Please note that Supporting Information files do not need this step.*

---

## [Author Response · Author response to Decision Letter 0]

13 Apr 2023

1. Please ensure that your manuscript meets PLOS ONE's style requirements

Those have been checked

2. In your Methods section, please provide additional information regarding the permits you obtained for the work.

The permit number has been added

3. We note that you have included the phrase “data not shown” in your manuscript.

It was badly used. I just did not display any figure for it, but the data are obviously accessible at the indicated figshare link. I deleted this sentence to avoid any misunderstanding, and just indicated the test results.

Response to reviewer: 

We made all the demanded changes in wording and clarified our point of view.

Title:

In the absence of extensive initial training, cleaner wrasse Labroides dimidiatus fail a transitive inference task. 

Reviewer #1: I believe this is a very good paper, well-written, clear, with properly analyzed results. It seems to me that the authors should stress more perhaps that fish showed a failure of training more than a failure of transfer (inference), i.e. that the latter is mostly the outcome of the former. At any event, I believe this paper can be published as it stands.

Thank you very much for this positive assessment. We agree that the title can be misleading as our paper is more a highlight of the biases of this TI task and a discussion on the methodological comparisons between our and Hotta et al. study. In light of the comments by referee 2, we also highlight more the importance of failure after the initial training in which they all managed to learn the different pairs.

Reviewer #2: The authors set out to test transitive inference (TI) in cleaner wrasse which are known for their social cognitive complexity. They reason the task is not ecologically relevant, so even though the fish are smart in other contexts, this particular task is irrelevant to them. The reasoning is sound with the exception of a previous paper on the same species and methodology which not only found evidence of TI but also justified why they expect this species to be capable of doing it on ecological grounds. So which expectation is true?? Moreover, TI is typically examined in a social context (agonistic interactions) but here the methodology is a foraging task. It is not immediately apparent if it is appropriate to expect that TI is transferable across contexts. At least some of the introduction needs to address “the cleaner fish in the closet”. Lastly, the test seems to be set up in a discrete way, whereas my intuition says that TI would best work with continuous variables.

We agree that we did not explain well the issue of ecological validity. In the new version, we quote what the Japanese team used as an argument. We disagree with them. We guess it is up to the reader to decide whether their logic or our logic is more parsimonious. They argue with the hierarchy within harems, where new members can use TI to infer the hierarchy without seeing each pair fighting. We think that this is not a good argument because cleaners (as other haremic sex changers) have a size-based hierarchy. So size alone predicts position within the hierarchy. A > B > C is absolute knowledge, no transitive inference is needed (A is always the “winner” and C is always the “looser”). The same logic holds for clients and their value as a function of body size. This contrasts with Grosenick et al. 2007 and their work on an African cichlid fish, where the authors used translocation to create different hierarchies with the very same fish. That was a brilliant design, and TI was absolutely necessary to get it right. Also note that cleaner harem size is rather small compared to the shoaling cichlids, as well as brook trout used by White and Gowan 2013, which can live in groups up to 27 individuals organized in a hierarchy impacted by different factors so as fish must adapt regularly.

Regarding our foraging paradigm (and the discrete reward scheme), we note that the majority of studies on TI have used it before us, including the Japanese team. We added a part just before the “reinforcement history” paragraph in the discussion to introduce the possibility that cleaners could potentially succeed in a social TI paradigm. It is true that the discrete side of could be a problem as the relationship must be continuous (e.g. stronger, bigger ..). We have it in our discussion (section “transitive inference test”) but we did not intend to discuss it more as multiple papers already discussed this problem (see Allen 2006, Guez and Audley 2013 for instance). Even if the relationship between the objects is not transitive (or let’s say, continuous), the response can still be transitive (see the reviews) Importantly, there are many positive results with this approach, including Hotta et al. 2020. Thus, our negative results are meaningful for a comparison.

One extra point that we now highlight in the methods section: our study was conducted in parallel with the Japanese group: we planned the study and collected the data without knowing about their study. Hence, the entire logic in the introduction is the logic we used to plan our study. We did not conduct the experiments to disprove them. We are just slower in publishing… to be precise, they submitted a first version to the journal on 23.3.20; we finished data collection on 15.3.20.

The authors seem to be claiming that they found no evidence for TI, but this is not really a true picture of the results. In fact, the fish basically failed the pre-training prerequisites and should never have been tested in the TI test at all. In other words, they were destined to fail. This is particularly problematic because the middle training pairs which are essential to the ultimate TI test were simply not learned above random. The authors mention this in the discussion, but ultimately it’s a fatal problem.

We adjusted the title to address this comment. However, it is incorrect to say that cleaners failed the pre-training prerequisites. They all learned to prefer the rewarding plate in all 4 plate combinations. Thus, we could have run the TI tests without any further testing. We interspersed the TI trials into a sequence of trials because we wanted a) have several trials for each individual on AE and BD to increase statistical power and mixing those trials with training ones seemed a good tool to lower chances of learning, and b) to look at performance in the 4 plate combinations when they are mixed. Those trials are the ones for which the data are presented. Note that it was a surprise to us that cleaners did not remember above chance the middle pairs. As numerously reported in the literature, we were expecting lower success (although the different order of learning groups should have flattened the curve a little) but the “not above chance” part was not expected. Only because of this design are we able to explain why the cleaners failed at the BD trials. What we explain better now is that they fail at the TI TASK, which doesn’t necessarily mean that they lack the cognitive process (explanation added just before the “reinforcement history” paragraph in the discussion). However, if you cannot memorize 4 relationships, then there is a memory constraint that prevents them from applying any TI even if they would in principle have the process in their repertoire..

In addition, our paper ultimately focuses on the why of the failure, mainly the biases leading to failure in the test of the training pairs and methodological comparisons with Hotta et al, which bring a better understanding of the cleaner wrasse cognition and of potential methodological improvements to be made in general. This is clarified by a change in the title.

We do more than mentioning this problem as the bigger part of the discussion focuses on this particular failure. The end of the first paragraph of the discussion is now clarifying the initial difference in our hypothesis so that it is easier for the reader to keep it in mind. 

I did not find the motivation for this study to be terribly convincing. It is just not clean. One paper says TI is to be expected given the fish’s ecology. Here the authors say its not. Would it not be simpler to just revisit Hotta and say you’ve greatly increased the sample size? That sounds like a better justification to me.

This was initially not based on Hotta et al. as the paper came out after we had collected the data. So saying this would be wrong and misleading, especially that there is this entire part of the discussion summarizing the differences between both methodologies. We adapted to their publication and compared both our studies to also allow more debate and discussion about the TI paradigm we used. We also explain why we think the ecology predicts negative results. We do not believe that the Hotta et al. justification is relevant. But their paper has been published and cannot be rewritten anymore. It does not necessarily imply that they are right and we are wrong. Also, the methods differed substantially, so talking about a revisit would be a little misleading.

Abstract: It is a bit odd that the opening statement suggest that you do no expect to find that cleaner wrasse are capable of AI given their ecology but your last sentence says there is already evidence of TI in this species. This is a little confusing.

We have to agree that it is a bit confusing but that is what empirical research can sometimes be about. As explained above, we follow the initial motivation to conduct the study. The abstract is too short to explain the differences between the 2 studies. In our view, the biggest methodological problem of the other study is that fish were also trained on AC and CE pairs. Nobody has done this before. In our view, this gives extra information about the order. It fixes ACE, and makes B and D embedded in different triangles. In summary our abstract says: “we don’t expect TI, we don’t find it, but as another study says yes, we discuss why there might be diverging results”. We do not believe this is misleading, but more of a scientific debate. 

L67: I guess the question here is, does it make sense to test TI outside of social hierarchies? The other cleaner fish paper did as I recall but the other fish ones used agonistic interactions.

The referee raises one of the most fundamental questions in cognition research, i.e. whether experimental designs should be rather abstract or capture as much ecological validity as possible. We think a combination of both types would be most powerful. If TI were a general tool, then subjects should be able to apply TI also in abstract tasks like the one we used. As noted earlier, most experiments on TI used the design we used in our study, including the Japanese colleagues.

L70: This is fine, but 2 of the 3 studies done on fish support the ecological explanation. The other doesn’t, although the introduction of Hotta et al 2020 claims their task is ecologically relevant. Here you claim it isn’t. So who to believe??

This has now been clarified in different places in the paper, and it remains up to the reader to decide which justification makes sense for him/her. Picking up on the referee’s comment regarding L67: the experimental design for cleaners is not ecologically relevant. Therefore, independently of whether or not cleaners could benefit from using TI in their social environment, the task used by the Japanese colleagues and us tests for an abstract presence of TI.

L87: So here you are suggesting they don’t need TI because size is a true predictor of hierarchy position. Do you know if the hierarchy in your model is truly linear? How do you then explain the previous result which used pretty much the same approach? Moreover, im not sure you can transpose TI from a social context to a feeding context. This is an interesting question in its own right.

This comment goes in two directions and picks up earlier points. An honest answer is that nobody has precise data on the strictness of the correlation between size and dominance. But we are talking about ever growing females in a sex changing species, living in harems of relatively small number of individuals. The probability of joining a harem that includes five females of almost similar size (Thus that size alone does not allow a prediction of the hierarchy) seems to be almost zero to us. Regarding the second part: indeed, both studies rely on cleaners being able to apply TI to an abstract scenario in order to succeed. 

L135: I have niggling doubts about this task. In my mind TI would be particularly useful for determining where things lie on a sliding scale, A is bigger than B which is bigger than C. But here the procedure is a little bit abstract. In some cases B has a reward (B+C1), in others it doesn’t (A+B-). It’s quite complicated.

You are absolutely right, and we also put this in our discussion (see “transitive inference task” section). The main point is that a binary paradigm is, by definition, not representing a transitive (or sliding scale, such as bigger, stronger etc..) relationship. But as we also explained, there are more complications to this. Mainly absolute knowledge, which is then also a major problem to talk about transitive inference. We chose a commonly used paradigm, but we think that some other research group would be more relevant to investigate this question in detail (it actually already being debated in the literature, so the question is absolutely known and raised already)

L137: in previous iterations of this test the “non preferred” plate would have flake food. Why no reward as apposed to a non-preferred award?

There are multiple complications in using preferred vs non-preferred food items. The individual preferences / tolerance are not equal and fix so we would need to be able to adapt quickly to avoid biasing the learning. The food/no food is easier to keep unchanged for the entire duration of this experiment (the data collection lasted for a bit less than 40 days for the slower individuals). And also, again, we used a standardized method to allow comparison. I hope the answer is relevant as I am not sure which iterations you are talking about. Hotta et. al did not use flake either.

L228L sooo the fish failed to learn the middle associations? Is there any point in testing for TI given this result? Especially since your test involved contrasts with D

See the method sections. Those data were collected simultaneously to the TI data so it was unknown. In addition, they all passed the training so we actually did not have any obligation to analyse those. We just wanted to make sure we had all the possible informations to understand the process. As we point out now very clearly, cleaners fail the TASK. Which is not the same as the conclusion that they lack TI as a mechanism. Indeed, they may fail the task simply because they cannot remember four combinations.

L235: Again this is a problem since the outcome of the TI will be dependent on what came before it.

See reply above.

L281: Yeah im not sure you can sell this idea too hard given Hora et al’s results. What you could say is you have repeated that test with greater sample size. I think this would be a more convincing motivation for the study.

As explained above this would not be true; it is not a repeat study. Therefore, the training methods differ. And we are pretty confident that Hotta et.al had positive results because of their overtraining. 

L287: My feeling is that your training regime failed, so the test for TI was not valid. You must seriously consider this as the main reason for your negative finding.

It is presented as is in the discussion. You are right that the failure may be due to a lack of memory rather than a lack of TI. But then TI would become rather useless if memory limits prevent fish from using it.

L356 & L381: see my earlier comments about this being a discreet test as opposed to a continuous test.

See replies above.

L389: except they don’t because your training failed. This is likely a false negative.

Again, they all passed the learning. The fact that it does not remain in their memory goes in the same direction: they do not need it. Apparently, an intense and extensive training such as the one Hotta et al. used can overcome this, but we are then totally outside the frame of what TI is helpful for: diminishing the amount of exposure needed to get information. This is also now added in the second part on the discussion.

---

## [Decision Letter · Decision Letter 1]

19 Apr 2023

PONE-D-23-02374R1In the absence of extensive initial training, cleaner wrasse *Labroides dimidiatus* fail a transitive inference task.PLOS ONE

Dear Dr. Bonin,

Thank you for submitting your manuscript to PLOS ONE. After careful consideration, we feel that it has merit but does not fully meet PLOS ONE’s publication criteria as it currently stands. Therefore, we invite you to submit a revised version of the manuscript that addresses the points raised during the review process.

ACADEMIC EDITOR: Thank you for revising the original manuscript. The reviewer (who kindly agreed to do a second review following their suggested changes in round 1) and I feel you have now adequately addressed all major comments. We both recommend minor revisions - to do a further proof-read paying specific attention to grammar corrections (some of which have been highlighted by the reviewer), particularly of the new revised text.  

We look forward to receiving your revised manuscript.

Kind regards,

Rachael Miller (Harrison)

Academic Editor

PLOS ONE

Journal Requirements:

Reviewers' comments:

Reviewer's Responses to Questions

**Comments to the Author**

1. If the authors have adequately addressed your comments raised in a previous round of review and you feel that this manuscript is now acceptable for publication, you may indicate that here to bypass the “Comments to the Author” section, enter your conflict of interest statement in the “Confidential to Editor” section, and submit your "Accept" recommendation.

Reviewer #2: All comments have been addressed

2. Is the manuscript technically sound, and do the data support the conclusions?

Reviewer #2: Yes

3. Has the statistical analysis been performed appropriately and rigorously? 

Reviewer #2: Yes

4. Have the authors made all data underlying the findings in their manuscript fully available?

Reviewer #2: Yes

5. Is the manuscript presented in an intelligible fashion and written in standard English?

Reviewer #2: Yes

6. Review Comments to the Author

Reviewer #2: The authors have done a great job addressing my earlier comments. The MS could still use a proof read by a native English speaker to improve the grammar. I found multiple grammatical errors (see below) but there are others.

L38: I suggest rephrasing the final sentence of the abstract: Indeed, a parallel study on cleaner wrasse provided positive evidence for TI but was achieved following extensive training on the non-adjacent pairs which may have over-ridden the ecological relevance of the task.

L46: “stores”, not “stocks”

L57: insert bracket after “mice”

L57 & 311: brook trout (trout is singular and plural)

L82: 3.5

Ll314-5: delete “in their brain”

335: replace “who” with “that”. Delete “the”

7. PLOS authors have the option to publish the peer review history of their article (what does this mean?). If published, this will include your full peer review and any attached files.

Reviewer #2: **Yes: **Culum Brown

---

## [Author Response · Author response to Decision Letter 1]

1 Jun 2023

The rephrasing has been used as is in the new version of the manuscript. In addition, grammar and clarity have been re-checked by both an appropriated software and a native speaker.

---

## [Editor Report · Decision Letter 2]

5 Jun 2023

In the absence of extensive initial training, cleaner wrasse *Labroides dimidiatus* fail a transitive inference task.

PONE-D-23-02374R2

Dear Dr. Bonin,

We’re pleased to inform you that your manuscript has been judged scientifically suitable for publication and will be formally accepted for publication once it meets all outstanding technical requirements.

Kind regards,

Rachael Miller (Harrison)

Academic Editor

PLOS ONE

Additional Editor Comments (optional):

No further comments

Reviewers' comments:

None

---

## [Editor Report · Acceptance letter]

13 Jun 2023

PONE-D-23-02374R2 

In the absence of extensive initial training, cleaner wrasse *Labroides dimidiatus* fail a transitive inference task. 

Dear Dr. Bonin:

I'm pleased to inform you that your manuscript has been deemed suitable for publication in PLOS ONE. Congratulations! Your manuscript is now with our production department. 

Kind regards, 

on behalf of

Dr. Rachael Miller (Harrison) 

Academic Editor

PLOS ONE